# Treatment of Diabetes Nephropathy in Mice by Germinating Seeds of *Euryale ferox* through Improving Oxidative Stress

**DOI:** 10.3390/foods12040767

**Published:** 2023-02-09

**Authors:** Yani Wang, Huaibo Yuan, Yidi Wang

**Affiliations:** School of Food and Biological Engineering, Hefei University of Technology, Hefei 230009, China

**Keywords:** type 2 diabetes, oxidative stress, germination, *Gordon Euryale* seed

## Abstract

Diabetes can cause severe kidney disease. *Euryale ferox* seeds (*Gordon Euryale*) have known antioxidant, hypoglycemic, and renal protection effects. Methanol extracts of *Gordon Euryale* were produced from ungerminated and germinated seeds. The effect of germination on polyphenol and flavonoid content was investigated by Liquid chromatography-tandem mass spectrometry (LC-MS/MS) analysis. Three doses of ungerminated seed extract (EKE) and germinated seed extract (GEKE) were administered to diabetic mice by gavage to explore the treatment-dependent improvement of oxidative stress, metabolic disorder, and kidney disease. Seed germination led to a 1.7 times increase in total phenol content in the extract, and the flavonoid content was increased by 1.9 times. Germination greatly increased the contents of 29 polyphenols and 1 terpenoid. At the same dose, GEKE more strongly improved hyperglycemia, abnormal lipid metabolism, and renal tissue lesions (as confirmed by histology) in the diabetic mice than EKE did. In diabetic mice receiving treatment, kidney microalbunminuria (ALB), blood urea nitrogen (BUN), serum creatinine (Scr), malondialdehyde (MDA), and glutathione (GSH) were all decreased, while activity of catalase (CAT), superoxide dismutase (SOD), and serum total antioxidant capacity (T-AOC) were increased. Both EKE and GEKE can improve diabetes and kidney disease by improving hyperglycemia, oxidative stress, and kidney physiological indicators and regulating the Keap1/Nrf2/HO-1 and AMPK/mTOR pathways. However, in both pathways, GEKE is more effective. The purpose of this study was to explore the effects of GEKE and EKE treatment on antioxidant defense and metabolic capacity of diabetic animals. Germination provides a suitable strategy to improve the medicinal value of these natural plant-based products.

## 1. Introduction

The prominent feature of diabetes is that the blood sugar value is higher than the standard. It may be caused by insufficient insulin secretion, insulin resistance, or a disorder of protein, fat, and carbohydrate metabolism [1,2]. A total of 90% of diabetes cases are type 2 diabetes (T2D) [3]. T2D is often accompanied by a series of complications, such as kidney disease, retinopathy, neuropathy, and cardiovascular and cerebrovascular diseases. A common complication is diabetic nephropathy, caused by microangiopathy, with an incidence between 20 and 50%. It is also the main cause of death and disability of patients with diabetes [4,5]. Diabetic nephropathy is characterized by glomerular hypertrophy, albuminuria, tubulointerstitial fibrosis, and decreased renal filtration rate. It is a multifunctional degenerative syndrome [6]. Although the specific pathogenesis of diabetic nephropathy is still unclear, a key factor is abnormal glucose and lipid metabolism. Long-term exposure to glucose and lipid overloads can lead to oxidative stress and inflammation in renal cells, eventually leading to interstitial rupture, glomerular hyperfiltration, glomerular sclerosis, and even end-stage renal failure [7].

Hyperglycemia leads to oxidative stress and increased levels of reactive oxygen species (ROS), which leads to complications of diabetes [8]. In T2D, ROS overproduction can occur through several pathways, including a continuous formation of advanced glycosylation end products (AGEs), increased activation of protein kinase C, or overactive polyol pathway or hexosamine pathways [9,10]. Once the overproduction of ROS exceeds the antioxidant defense system, imbalanced oxidation reduction reactions produce oxidative stress conditions. Oxidative stress can lead to cell damage and physiological dysfunction through ROS-mediated oxidation of macromolecules such as DNA, proteins, and lipids [11,12,13]. Renal cell apoptosis caused by metabolic disorders leading to oxidative stress is believed to be involved in the development of diabetic nephropathy. It follows that inhibition of renal cell apoptosis by improving oxidative stress may provide a therapeutic target for this condition.

A key antioxidative signaling pathway involved in the antioxidant defense system is Keap1/Nrf2/HO-1. This pathway regulates the expression of various ROS through antioxidant factors including oxidase-plus 1 (HO-1), NAD(P)H:quinone oxidoreductase-1 (NQO1), catalase (CAT), total superoxide dismutase (T-SOD), and glycopeptin peroxidase (GSH-Px) [14]. Nuclear factor Erythrocyte 2 related factor-2 (Nrf2) regulates the expression of antioxidant response elements and is an important regulator of cell homeostasis. Kelch-like ECH-related protein 1 (Keap1) is the main inhibitor of Nrf2 expression. Under normal physiological circumstances, Nrf2 binds to Keap1 and is degraded by ubiquitination or by the proteasome. When Keap1 activity is inhibited, Nrf2 accumulates and translocates to the nucleus, where it induces transcription of a number of genes to activate the ROS defense system [15,16]. Nrf2 shares common stressors with AMP-activated kinase (AMPK) and the cellular responses resulting from their activation overlap. This suggests that AMPK and Nrf2 signaling may depend on each other and that they cooperate to re-regulate cellular homeostasis [17]. AMPK has been proved to regulate cell energy homeostasis and control redox balance and inflammation. In the context of the pathological mechanism of diabetic nephropathy, it has been pointed out that high glucose levels affect AMPK phosphorylation by inducing phosphorylation on Ser2448 and this activates mTOR, resulting in induced extracellular matrix accumulation in renal fibroblasts. The renal AMPK activity in diabetes is related to renal autophagy dysfunction and renal cell apoptosis. The decrease in AMPK activity changes the renal protein synthesis and mitochondrial function [18,19].

Plant ingredients can provide an attractive alternative to synthetic drugs to treat chronic diseases such as diabetes and this has been an ongoing research focus. Plant polyphenols and flavonoids are widely found in nature and have good anti-inflammatory, antioxidant, and metabolic regulatory functions. In particular, during the germination of plant seeds, material is transformed at low cost, providing an effective processing technology that can effectively reduce antinutritional factors present in the seeds (tannin, phytic acid, protease inhibitors) [20] while enriching beneficial bioactive components (GABA, vitamins, polyphenols, flavonoids, etc.). Germination promotes the degradation of the plant seed cell wall, releases phenolic substances, increases the content of phenolic acids and flavonoids, and improves the antioxidant activity of seeds [21,22].

Seeds of *Euryale ferox* (*EF*) (prickly waterlily, a member of the Nymphaeaceae family) are sold in dried form as *Gordon Euryale* (*GE*) and are used in traditional Chinese medicine to treat a variety of ailments including chronic diarrhea, kidney problems, spleen dysfunction, and heart muscle damage. The seeds are rich in lignans, tocopherols, cyclic peptides, cerebral glycosides, polyphenols, and flavonoids, and further contain glucosterols, alkanes, and lipids. Of particular interest are their phenolic compounds, as plant polyphenols can have antioxidant, anti-bacterial, and anti-viral effects. GE seeds have been shown to contain ferulic acid, kaenerol, quercetin, rutin, caffeic acid, gallic acid, and chlorogenic acid, with confirmed antioxidant activity in vitro [23].

At present, research on *GE* is mainly focused on the medicinal value of the dried seeds, but little research is available on the effect of seed germination on its efficacy. Here, we prepared extracts from *GE* seeds prior to germination (EKE) and following germination of the seeds (GEKE) and these extracts were used to treat streptozotocin (STZ)-induced type II diabetic mice. By comparing the treatment effects, the effect of germination of the seeds on the improvement of oxidative stress in diabetic kidneys was evaluated to confirm whether seed germination is beneficial for improving oxidative stress conditions.

## 2. Materials and Methods

### 2.1. Preparation of Gordon Euryale Kernel Extract (EKE) and Germinated Gordon Euryale Kernel Extract (GEKE)

*Gordon Euryale* seeds were purchased from Liu an Anhui Province Kangzhiyuan, a *GE* seed production facility. Half the seeds were vernalized at 4 °C for 60 days and then immersed in water at 25 °C for germination. Germinated seeds were used once the germs reached a length of 0.5 cm (after 15–20 days).

Ungerminated and germinated seeds were shelled and crushed and passed through a 200 mesh sieve to obtain powder. Extracts from these powders were produced by immersion extraction in 70% methanol [24] at a ratio of 1:10 (*w/v*), followed by ultrasonic treatment at 320 W for 1 h at 70 °C. They were centrifuged at 4 °C (12,000 rpm, 5 min), then the supernatants, called GEKE (from germinated seeds) or EKE (from ungerminated seeds), were stored at 4 °C to be used for further experiments.

### 2.2. Phytochemical Assays

#### 2.2.1. Quantification of Total Polyphenols

The spectrophotometric method based on Folin–Ciocalteu reagent [25] was used to determine total phenolic content (TPC). For this, 1.2 mL Folin-silica ceramic reagent was added to 0.3 mL of GEKE or EKE, respectively, and incubated for 1 min. Following the addition of 2.4 mL 10% sodium carbonate and water to a final volume of 10 mL, the mixture was incubated in the dark for 2 h at ambient temperature. Absorbance at 750 nm was measured with a UV775B spectrophotometer and a calibration curve based on gallic acid (0, 1, 2, 3, 4, 5, 6 μg/mL) served for quantification of total polyphenols. TPC content is expressed in mg of gallic acid equivalent (EAG) per gram of extract, and the linear regression equation of the standard curve is as follows: $Y=0.1566X-0.0247,{\;R}^{2}=0.9945$.

#### 2.2.2. Quantification of Flavonoids

The content of flavonoids in the extract was determined by aluminum chloride colorimetry. For this, 1 mL of EKE or GEKE was added to 0.3 mL 5% NaNO_2_, and following incubation for 6 min, 0.3 mL of 10% Al(NO_3_)_3_ was added. Following 6 min incubation, 4 mL 4% NaOH was added, and after further incubation for 15 min at room temperature, absorbance was determined at 510 nm. Quantification was based on a calibration curve of rutin solution (0, 10, 20, 30, 40, 50 μg/mL) ($Y=0.0143X-0.0027,{\;R}^{2}=0.9962$) and the amount of flavonoids was expressed in milligram rutin equivalents (ER) per g EKE or GEKE [26].

#### 2.2.3. Soluble Protein Content

The soluble protein content of the extracts was determined spectrophotometrically with Coomassie Brilliant Blue G-250 [27]. For this, colorant (0.1 g) was dissolved in 50 mL 90% ethanol containing 100 mL 85% phosphoric acid, and water was added to a final volume of 1 L. Of this, 5 mL was added to 1.0 mL EKE or GEKE, respectively. After shaking at room temperature for 5 min, the absorbance was measured at 595 nm. A calibration curve was made with bovine serum protein (BSA) in 0.9% NaCl (0, 10, 20, 30, 40, 60,80 μg/mL) ($Y=0.0128X-0.0598,{\;R}^{2}=0.9522)$.

### 2.3. LC-MS/MS Analysis

The extract was analyzed by liquid chromatography (LC) using UltiMate 3000 RS chromatograph (Thermo Fisher Scientific Co., Ltd., Shanghai, China). The chromatographic column (RP-C18, 1.8 μm, measuring 2.1 × 150 mm) was used with a flow rate of 300 μL/min. The column oven temperature was 35 °C and the autosampler was set at 10 °C. The injection volume was 5.00 μL. Solvents A (water/0.1% formic acid) and B (acetonitrile/0.1% acetic acid) were used at a constant flow rate (0.30 mL/min). A non-linear gradient was applied as follows: 2% B (0–1 min), 20% B (0–5 min), 50% B (5–10 min), 80% B (10–15 min), 95% B (15–20 min), 95% B (20–25 min), 2% B (25–26 min), 2% B (26–30 min).

Mass spectrometry (MS) analysis was carried out under the conditions of positive and negative ion switch scanning and full mass/DD-MS2 detection using a Q Active high-resolution mass spectrometer (Thermo Fisher Scientific Co., Ltd., Shanghai, China), equipped with an electric spray ionization source. Samples were analyzed with a scan range of M/Z 150–2000 in Fourier transform mass spectrometry (FTMS) (Thermo Fisher Scientific Co., Ltd., Shanghai, China) mode. The detection resolution of the sample was set to 70,000 (full mass), 17,500 (dd-MS2). Other setting parameters were as follows: 3.8 kV Positive Electrospray Voltage; 300 °C capillary temperature; argon (purity ≥99.999%) as collision gas, nitrogen (purity ≥99.999%) as sheath gas at 40 Arb and as auxiliary gas at a temperature of 350 °C. Data were collected over 30 min [28].

Standard package software CD2.1 (Thermo Fisher Co., Ltd., Shanghai, China ) was used to complete the initial sorting of data and results were queried against standard databases (mzCloud, mzVault, ChemSpider).

### 2.4. Murine Model of Diabetes

All animal experiments were approved by the Biomedical Ethics Committee of Hefei University of Technology(ethics number: HFUT20220820001). Ninety male ICR mice with body weight of 20 ± 2 g were selected and placed in the standard laboratory animal room at 23 ± 2 °C for 12 h diurnal cycle. After a week of adaptive feeding, the mice were given different feedings. Ten mice were selected randomly as the normal control (NC) group and these were fed with a normal chow diet throughout the experiment, while the other 80 mice were fed with a high-fat diet consisting of 5% egg yolk, 8% lard, 10% sugar, 76% regular feed, and 1% cholesterol for 4 weeks.

The mice were then fasted for 12 h, and the animals that had received the high-fat diet were given a one-time intraperitoneal injection of 100 mg/kg STZ (STZ in sodium citrate buffer, pH 4.2–4.5). Three days after STZ injection, the animals were deprived of water for 12 h. Blood samples were collected from the caudal vein and fasting blood glucose (FBG) was measured by a glucose meter. All 80 STZ-treated mice had blood glucose values ≥11.1 mmol/L accompanied by obesity and polyuria, confirming STZ-induced type 2 diabetes.

### 2.5. Experimental Setup

The normal control group that had not received STZ (NC mice), was continuously fed an ordinary diet without any treatment. The type 2 diabetic mice (as described above) were randomly divided into 8 groups of 10 mice each, and these received treatment for four weeks as summarized in Table 1. When indicated, chow contained 200 mg/kg bw per day metformin HCL. Mice in the low-, middle-, and high-dose groups were administered 100, 200, and 400 mg/kg bw per day of either EKE or GEKE suspension once a day for four weeks by gavage [29].

### 2.6. Determination of Biochemical Parameters

Body weight and fasting blood glucose were measured and recorded periodically during the experiment. On the last day of treatment, the mice were tested for oral glucose tolerance following 12 h of fasting. For this, the mice were gavaged with 2.0 g/kg bw glucose solution (400 mg/mL), and the blood glucose levels were measured at 0 h, 0.5 h, 1 h, 1.5 h, and 2 h [30]. The area under the curve (*AUC*) of glucose levels was calculated as follows:$$AUC=\left(BG_{0\min}+BG_{30\min}\right)\times 0.5h\times 0.5+\left(BG_{30\min}+BG_{120\min}\right)\times 1.5h\times 0.5$$

At the end of treatment, the cervical vertebra of the mice was dislocated fatally. Eyeball blood was collected for 15 min and centrifuged (4 °C, 3000 rpm), and the serum supernatant was stored at −80 °C for subsequent tests. The surgical operation involved the removal of kidneys, spleen, and liver, and 0.1 g of tissue of each organ was homogenized in 1 mL PBS (PH 7.2–7.4) in an ice bath and centrifuged. These homogenate supernatants were stored at −80 °C for subsequent tests.

Visceral organ indices of liver, spleen, and kidney tissue were determined as organ weight (g)/body weight (g).

The enzyme activities of catalase (CAT), malondialdehyde (MDA), superoxide dismutase (SOD), and glutathione (GSH) in the supernatant of kidney homogenate (Beijing Sun God Technology Co., LTD, Beijing, China) were determined with a commercial kit. Serum lipid content was determined for total cholesterol (TC), LDL-cholesterol (LDL-C), HDL-cholesterol (HDL-C), and triglycerides (TG) using standard kits and a serum lipid index was calculated [31]. Total antioxidant capacity (T-AOC) was also determined (Elabscience Biotechnology Co., Ltd., Wuhan, China), as were Scr, BUN, and ALB levels, detected by standard kits (Jiancheng Co., Ltd., Nanjing, China).

### 2.7. Histological Examinations

Kidney tissue was fixed in 4% paraformaldehyde for 24 h and embedded in paraffin. Tissue slices of 4 mm thickness were prepared and stained with hematoxylin-eosin (H&E) [32] and Masson’s trichrome. The preparations were examined using a fully automated digital pathology section scanner and visually inspected with CaseViewer software 2.4.0 (Seville, Hefei, China).

### 2.8. Gene Expression Analysis by RT-PCR

Total RNA was extracted from mouse kidneys using an animal total RNA isolation kit (Shanghai Sangong Co., LTD., Shanghai, China). Sense and antisense primers (designed using the Primer Premier(Shanghai Sangong Co., LTD., Shanghai, China) 5 tool) for the genes coding for Keap1, Nrf2, HO-1, mTOR, AMPK, and β-actin (standard control) are shown in Table 2. Synthesis of cDNA by reverse transcriptase was performed with the M-mulv First Strand cDNA Synthesis kit (Sangon Co., Ltd., Shanghai, China) and quantitative PCR amplification was done using Taq PCR Master Mix (Sangon Co., Ltd., Shanghai, China) [33]. 

### 2.9. Protein Expression

The protein expression levels of Nrf2, HO-1, Keap1, p-mTOR, and p-AMPK genes in the kidney tissue were measured by corresponding ELISA kits (Jiangsu Enzyme Immunity Co. LTD., Nanjing, China). Nrf2, HO-1, Keap1, p-mTOR, and p-AMPK ELISA kits used in the experiment are all 96T specifications (Jiangsu Enzyme Immunity Co. LTD.).

### 2.10. Statistical Analysis

All measurements were repeated 3 times and expressed as mean ± standard deviation (SD). IBM SPSS 26.0 software (Suzhou Mairuan Network Co., Ltd., Suzhou, China) was used for one-way analysis of variance (ANOVA) to determine whether the differences were statistically significant. Dunnett’s multiple range test was performed on mean values and *p* < 0.05 was considered statistically significant.

## 3. Results

### 3.1. Total Polyphenol and Flavonoid Content

The extracts of non-germinated *Gordon Euryale* seeds (EKE) and of germinated seeds (GEKE) were analyzed for total phenolic contents. EKE contained 3.23 ± 0.1 mg EAG/g, which was 1.7 times lower than the content of GEKE, for which 5.503 ± 0.26 mg EAG/g was determined. Likewise, the content of flavonoids was only 2.357 ± 0.17 mg ER/g for EKE but nearly double, 4.443 ± 0.06 mg ER/g, for GEKE. Thus, germination of the seeds increased the levels of both total phenolics and of flavonoids.

### 3.2. LC-MS/MS Analysis of EKE and GEKE

The polyphenols and terpenoids present in GEKE and EKE that could be identified by LC-MS/MS are shown in Table 3. Increased content of 29 polyphenols and 1 terpenoid in the germinated seed extract could be demonstrated, while germination reduced the contents of 9 other polyphenols and 1 terpenoid. Before germination, 18 phenolics and 1 terpenoid were almost absent but could be detected in the extracts of germinated seeds. These included gallic acid, gentisic acid, and caffeic acid, as well as 2-hydrohenylalxypanine, L-tyrosine methyl ester, a glucopyranose derivate, cycloolivil, matairesinol, 3’,4’-dihydroxyphenylacetone, hydromorphinol, and estriol, amongst others (Table 3).

### 3.3. Body Weight and Visceral Indexes

STZ-induced diabetic mice were treated with EKE or GEKE at three concentrations for four weeks (Table 1). Because diabetes is closely associated with obesity, the weight of the mice was recorded during treatment (Figure 1). The weight of the STZ-treated mice had increased compared to the NC control. After 4 weeks of treatment, the body weight of NC, PC, GEKEM, GEKEH, and EKEH groups was lower than that of the diabetes control group (*p* < 0.01). The body weight of animals in the PC, GEKEM, and GEKEH groups after treatment was even lower than before treatment. In addition, the body weight of the GEKEL, EKEL, and EKEM groups was significantly reduced compared to the DC group (*p* < 0.05). We observed that the GEKE group was performing significantly better than the EKE group in terms of anti-obesity symptoms of type 2 diabetes. The weight of the GEKEM group was similar to that of the NC group after treatment. The GEKEH group resulted in outstanding improvement effects, and the weight loss effect of these animals was even better than the metformin hydrochloride treatment group (PC group).

After the experiment, the mice were dissected and the visceral index of the kidney and liver was determined. Table 4 summarizes the changes in these visceral indices. The kidney index of animals in the NC, GKEKM, GEKEH, and EKEH groups was much lower than that of the DC group (*p* < 0.01), while that index was less reduced but still significant in the NC, PC, GKEKL, and EKEM groups (*p* < 0.05). For liver tissue, the visceral index of all treatment groups and of the normal group was significantly lower than that of the DC group (*p* < 0.01). Lastly, the spleen index was much lower for NC, GEKEM, and GEKEH animals compared to the DC group (*p* < 0.01), with weaker but still significant reductions in the PC, GEKEL, and EKEH groups (*p* < 0.05). As can be seen from the specific values in Table 3, the organ indices of the GEKE group were lower than those of EKE group under the same dose. In addition, the effect of low-dose GEKE on inhibiting organ hypertrophy performed better than metformin (200 mg/kg bw) administration.

### 3.4. Fasting Blood Glucose (FBG) and Oral Glucose Tolerance Test (OGTT)

Figure 2 shows the changes in blood sugar in mice after treatment. Obviously, the FBG of T2D mice after SZT induction and high-caloric feeding was significantly higher than that of normal mice (*p* < 0.01). At the end of the treatment, during which the animals were fed with normal chow, the FBG had strongly decreased in the NC, PC, GEKEM, GEKEH, and EKEH groups compared to DC group (*p* < 0.01), and a weaker but significant decrease in FBG was noted in GEKEL and EKEM (*p* < 0.05). The only treatment that did not result in a significant reduction in FBG compared to DC was EKEL in which the animals received the lowest dose of the extract produced with non-germinated seeds (*p* > 0.05).

Four weeks after continuous treatment, glucose tolerance tests were performed following 12 h of fasting (Table 5). The results of this OGTT showed that the blood glucose of each group gradually decreased after reaching the maximum at 30 or 60 min. Except for the NC mice that returned to a normal range after 120 min, the blood glucose values of the other groups remained elevated at 120 min compared to 0 min, to varying degrees. At 120 min, glucose values in the GEKEH animals were significantly lower from those in the DC group (*p* < 0.01), with weaker effects for the GEKEM, EKEM, and EKEH groups (*p* < 0.05). The area under the curve (AUC) was determined for each group. The AUC of DC was strongly increased compared to NC (*p* < 0.01), and the AUC of the PC and GEKEM groups was significantly lower than that of the DC group (*p* < 0.05). Even stronger decreases in the AUC were observed in the GEKEH and EKEH groups compared to DC (*p* < 0.01). The AUC of the GEKEL, EKEL, and EKEM groups was not significantly different compared with the DC group (*p* > 0.05).

According to Figure 2 and Table 5, both the EKE group and the GEKE group showed dose-dependent improvement in hyperglycemia symptoms. The higher the dose of the same drug, the better the hypoglycemic ability. The therapeutic effect of the EKE treatment was not as good as that of GEKE, but the therapeutic effect of EKEH was similar to that of GEKEM, and the hypoglycemic effect of GEKEH treatment was far better than that of EKEH.

### 3.5. Serum Biochemical Parameters

Diabetes mellitus is characterized by dyslipidemia, especially hypertriglyceridemia and low HDL-C levels, accompanied by a reduction in TC and LDL-C. Both diabetes and dyslipidemia increase the risk of atherosclerotic vascular disease [34]. The blood of the sacrificed animals was used to determine a series of blood lipid levels (Table 6). The total cholesterol (TC) value in GEKEM, GEKEH, and EKEH groups was much lower than that in the DC group (*p* < 0.01). The triglyceride (TG) level of the EKEL group was close to that of PC and not significantly different to that of DC (*p* > 0.05). TG values in the GEKEL group were slightly lower compared to DC (*p* < 0.05), whereas the TG values in the other treatment groups were more strongly decreased (*p* < 0.01). LDL-C values in the PC, GEKEM, and EKEM groups were all significantly lower than DC (*p* < 0.05), and these values were again lower with strong significance in the GEKEH and EKEH groups (*p* < 0.01). Lastly, HDL-C values in the PC, GEKEM, and EKEH groups were significantly higher than in the DC group (*p* < 0.05), while for the GEKEH group these levels were significantly lower than in the DC group (*p* < 0.01). From these data, it can be concluded that there is a therapeutic effect of both EKE and GEKE, although the extract of germinated seeds results in stronger effects, with a certain dose dependence.

### 3.6. Renal Indicators

Table 7 shows the renal function index in each group of mice. It can be seen that the serum ALB, BUN, and Scr values in the GEKEM and GEKEH groups were all significantly different from those in the DC group (*p* < 0.01), whereas differences in the EKEL, EKEM, and GEKEL groups were either minor or not significant. As for the EKEH group, the BUN level was significantly higher than that of the DC group (*p* < 0.01). There was no significant difference between the GEKE and EKE groups in improving BUN secretion level. However, according to the data, GEKE treatment performed significantly better than EKE at the same dose, in terms of improving the levels of ALB and Scr. The effect of reducing ALB and Scr in the GEKEL group was similar to that in the PC group, and the therapeutic effect of GEKEM was such that values now resembled the NC group.

### 3.7. Oxidative Stress Indicators

Figure 3 shows renal oxidative stress parameters and serum total antioxidant levels of the treated mice. Comparing the DC and NC groups, it can be seen that the activity levels of CAT and T-AOC in diabetic mice had decreased strongly (*p* < 0.01), and the activity of SOD had decreased significantly (*p* < 0.05), while the activity of GSH had increased even more strongly (*p* < 0.01), accompanied by an increase in the activity of MDA (*p* < 0.05). Following intervention treatment, the levels of SOD, MDA, CAT, GSH, and T-AOC in the GEKEH group were all different, with strong significance compared to the DC group (*p* < 0.01). Likewise, CAT and T-AOC levels were strongly elevated in the GEKEM group (*p* < 0.01) and the latter was similarly elevated in the GEKEL group (*p* < 0.01). Treatment with extract from non-germinated seeds resulted in strong differences in levels of MDA, CAT, GSH, and T-AOC for EKEM compared to DC (*p* < 0.01). The other indexes in these groups produced low or not significant differences compared with the DC group. In conclusion, GEKE treatment can improve the antioxidant capacity of the body more strongly than EKE under the same dose condition. This can be intuitively seen from the level of T-AOC; the T-AOC level of the GEKEM group was close to that of the NC group and was significantly better than that of the GEKEH group.

### 3.8. Pathological Changes in the Kidney

To study the effects of the two tested extracts on renal cell structure in diabetic mice, paraffin sections of mouse kidneys were stained with H&E and Masson and the renal pathological status in each group was evaluated. As shown in Figure 4, the kidney structure of animals from the NC group was regular, with normal glomeruli and renal tubules that were evenly arranged and an absence of fat particle deposition or mesangial matrix hyperplasia. In diabetic control animals, DC, as in the EKEL and GEKPL groups, the renal tissue structure was disordered, with presence of glomerular atrophy, renal tubule dilatation, renal interstitial edema, partial renal interstitial fibrosis, and cell atrophy and shedding. However, compared with the DC group, the renal pathological signs of the GEKEM, GEKEH, EKEM, and EKEH groups were significantly reduced, and in particular, the renal pathological conditions of the GEKEH and EKEH groups were greatly improved. The renal structure of the GEKEH group was most similar to that of the NC group. This clearly demonstrates that EKE and GEKE treatment had renal protective effects. In addition, the high-dose GEKE treatment resulted in the best renal protection.

### 3.9. Renal Gene mRNA Levels and Protein Expression

To explore the molecular mechanism of EKE and GEKE treatment on nephropathy in diabetic mice, the transcript levels of various genes were determined in kidney homogenate. As shown in Figure 5, the RT-PCR results show that the expression of important genes involved in the AMPK/mTOR and Keap1/Nrf2/HO-1 signaling pathways, keap1, nrf2, and ampk, was significantly increased in the DC group compared with NC, while the expression of transcripts coding for HO-1 and mTOR was significantly decreased in DC. Following intervention with different doses of GEKE or EKE, these gene expression levels responded to different degrees. The mRNA levels of transcripts for Nrf2, HO-1, and mTOR were increased as a result of treatment, while the transcription levels of Keap1 and AMPK were decreased. The gene expression level gradually shifted towards that of the NC group with an increase in treatment dose.

Figure 6 shows the protein expression levels of key enzymes in kidney samples. HO-1 levels decreased significantly in the DC group compared with NC (Figure 6A). This protein was increased compared to DC following GEKEL and EKEH treatment (*p* < 0.05), and more strongly so in GEKEM and GEKEH (*p* < 0.01), where levels were close to that in the NC group. Figure 6B shows that Keap1 levels were significantly increased in the DC group compared with the NC group. Treatment reduced the levels of Keap1 in each group. Among the observed changes, the levels of Keap1 in PC, GEKEH, and EKEH groups now resembled those of the NC group, with significant differences compared with the DC group (*p* < 0.01). Figure 6C illustrates that Nrf2 levels increased in the DC group compared with the NC group. Compared with DC, the level of Nrf2 in GEKE and EKE for all three dose groups increased, with the strongest increase in the GEKEH group (*p* < 0.01). Figure 6D summarizes the *p*-mTOR levels. This protein increased significantly in the DC group compared with the NC group, and except for the treatment with EKEL, the levels of p-mTOR in the other groups were significantly lower than in DC (*p* < 0.01). Lastly, Figure 6E shows that p-AMPK levels decreased in the DC group compared with the NC group and that, compared to DC, GEKEL, EKEM, and EKEH treatment significantly improved the p-AMPK, with even stronger effects for GEKEM and GEKEH.

According to the mRNA and protein expression results of key renal oxidative stress genes, the regulation effect of high-dose EKE on HO-1 and p-mTOR levels is similar to that of low-dose GEKE. The effect of high-dose EKE on regulating Keap1 and Nrf2 levels was similar to that of medium-dose GEKE. The effect of low-dose GEKE on the regulation of p-AMPK level was significantly better than that of high-dose EKE. It can be seen that GEKE has a more stable effect in regulating the expression of the AMPK/mTOR and Keap1/Nrf2/HO-1 signaling pathways, and low-dose GEKE is enough to make a difference with the DC group.

## 4. Discussion

The results of this study show that the germination and cultivation of Euryale seeds can improve its medicinal value of antioxidative stress and renal disease by increasing the content and richness of phenolic substances. The used extracts were analyzed by LC-MS/MS, which identified that germination of the seeds increased the content of 29 identified phenolic substances and 1 terpenoid, among which 18 phenolic substances and 1 terpenoid newly emerged that were not detectable in ungerminated seeds. Amongst other differences, germinated *GE* seeds contain much more caffeic acid, gentian acid, and a number of other polyphenols compared to ungerminated seeds. 

When diabetic mice were treated with these extracts, their physiological indexes improved, in particular following GEKE administration, which greatly improved the symptoms of obesity, hyperglycemia, lipid metabolism disorder, and diabetic nephropathy. GEKE did this more effectively compared to EKE at the same dose. In addition, the therapeutic effects of both EKE and GEKE extracts were dose-dependent. Among them, the oxidative stress index, kidney index, and kidney tissue status following GEKEH treatment were closest to values obtained for normal control animals, and the difference was statistically significant compared to the diabetic control group. Following detection of certain indicators (ALB, Scr, SOD, CAT, T-AOC), we observed that the therapeutic effect of GEKE at a low dose (GEKEL) was comparable to that of metformin treatment, while EKE only reached this at a high dose (EKEH). Thus, even low doses of the germinated *GE* seed kernel extract already have a good therapeutic effect on diabetic mice. Germination of the seeds offers a low-cost processing method to improve the content of active substances in *GE* seeds and thus improves their anti-diabetic oxidative stress effect.

Polyphenols have the ability to remove ROS from cells [35]. These components have potential in the treatment of diabetes. For instance, gallic acid can inhibit the reduction in catalase and glutathione S-transferase activities as well as vitamin C levels in the liver of diabetic rats, and it has a protective effect on renal tissue injury caused by oxidative stress in diabetic patients [36]. Gentian acid, a byproduct of tyrosine and benzoic acid metabolism, has anti-inflammatory, rheumatic, and antioxidant biological activities [37]. Caffeic acid has strong antioxidant and anti-inflammatory activities [38]. Fustine is a flavonoid that has been reported to have hypoglycemic, antioxidant, anti-arthritic, anti-obesity, and anti-cancer effects; it can reduce the level of MDA and increases the levels of GSH, SOD, and CAT in diabetic rats [39]. Our hypothesis, that germination of the *GE* seeds might improve their medicinal value in terms of anti-inflammatory, antioxidant, hypoglycemic, and metabolic regulation, was confirmed, and this was related to the observed enrichment of polyphenol active substances such as gallic acid, gentian acid, caffeic acid, fustine, L-tyrosine methyl ester, and others [40,41].

Excessive oxidative stress and long-term hyperglycemia caused by diabetes can directly damage renal tissue, leading to different degrees of pathological renal injury and changing renal hemodynamics and glomerular permeability [42,43,44]. In the treatment of diabetic nephropathy, targets of interest are genes related to antioxidation. The antioxidative signaling pathway Keap1/Nrf2/HO-1 regulates ROS levels via antioxidant genes, including HO-1, CAT, GSH-px, and T-SOD. This pathway plays a pivotal role in antioxidant defense against diabetic nephropathy. In addition, Nrf2 is a key transcription regulator for antioxidant and detoxification enzymes. The activation of Nrf2 can prevent oxidative stress and inhibit the development of inflammation, thus improving diabetic kidney disease. It has been shown that Nrf2 inhibition leads to AMPK activation and mTOR pathway inhibition by regulating ATP consumption [45]. The AMPK/mTOR signaling pathway is also closely involved in diabetic nephropathy. At present, plant components such as quercetin, genistein, and berberine have been shown to increase AMPK activation in order to regulate oxidative stress and inflammation. When applied to diabetic nephropathy, these could prevent renal injury and inhibit podocyte injury and high-glucose-induced apoptosis [46].

By analyzing transcripts and protein activity of key components of the Keap1/Nrf2/HO-1 and AMPK/mTOR pathways, we further collected evidence of the mechanism by which oral polyphenol administration can significantly improve diabetes and its complications. We found that the expression of Nrf2, Keap1, and p-mTOR in the kidneys of diabetic mice was increased without treatment, indicative of an adaptive response to oxidative stress. The observed decrease in HO-1 and p-AMPK levels in the DC group may be caused by excessive consumption of the renal oxidative stress response, leading to further enhancement of oxidative stress. After treatment with the GEKE and EKE extracts, the expressions of Nrf2, HO-1, and p-AMPK in the kidneys of diabetic mice were significantly increased, while the expressions of p-mTOR and Keap1 were significantly decreased. At the same time, SOD and CAT activity was increased, while MDA and GSH were decreased, and the antioxidant ability was enhanced. The serum levels of ALB, BUN, and Scr were decreased as a result of treatment, and the renal-related lipid indexes were improved, with decreased levels of TC, TG, and LDL-C and increased levels of HDL-C, so that the diabetic, abnormal lipid metabolism was vastly improved. In combination, we consider that these results strongly indicate that EKE and GEKE treatment exerts a beneficial activity via antioxidative stress and renal protection by regulating Keap1/Nrf2/HO-1 and AMPK/mTOR signaling pathways. As the activity level of indicators for these pathways suggests, GEKE displayed a stronger therapeutic effect than EKE.

## 5. Conclusions

Through intragastric administration of EKE and GEKE in mice, the physiological indexes of mice were improved, and there was a certain correlation with the intragastric dose. Among them, the therapeutic effect of GEKE was better. The structure analysis of the two extracts showed that GEKE contained more caffeic acid, gentic acid, and other effective substances. It can be concluded that germination of the *GE* seeds increases the content of beneficial and effective polyphenols, so that germination improves the medicinal value to treat the oxidative stress symptoms of diabetes. This conclusion was based on experiments with methanol extracts that were richer in polyphenols following germination of the seeds. In mouse experiments, the extracts were shown to improve antioxidant indexes and renal indexes. GEKE can effectively improve the antioxidative stress ability and renal protection ability of diabetic animals by clearing excessive ROS and activating Keap1/Nrf2/HO-1 and AMPK/mTOR signaling pathways. The purpose of this study is to provide a new way to improve the medicinal value of *EF* seeds.

## Figures and Tables

**Figure 1 foods-12-00767-f001:**
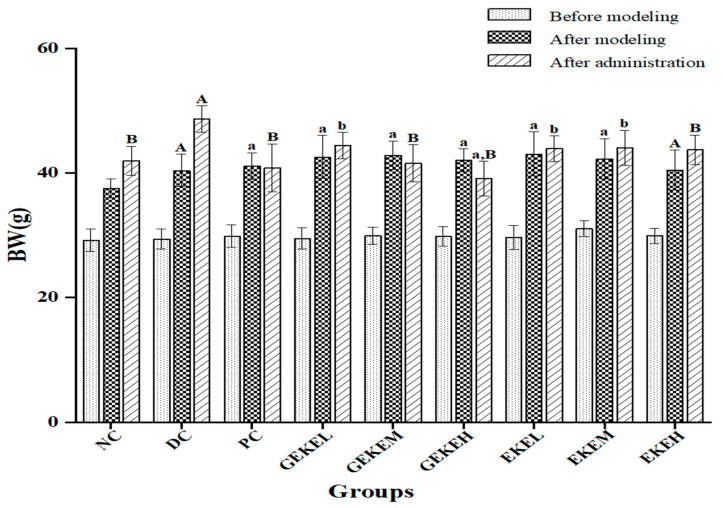
Effect of *Gordon Euryale* seed kernel extract (EKE) and germinated *Gordon Euryale* seed kernel extract (GEKE) on the body weight of diabetic mice. A total of 10 per group(n = 10); data are presented as mean ± SD values. A ( *p* < 0.01) and a (*p* < 0.05) vs. the normal control; B (*p* < 0.01) and b (*p* < 0.05) vs. the diabetes control.

**Figure 2 foods-12-00767-f002:**
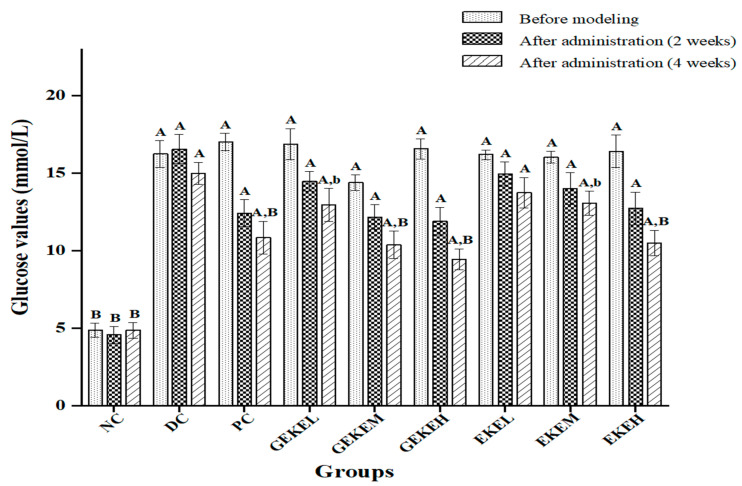
Effect of *Gordon Euryale* seed kernel extract (EKE) and germinated *Gordon Euryale* seed kernel extract (GEKE) on blood glucose in type 2 diabetic mice (n = 10 per group). Data are presented as mean ± standard deviation (SD) values. A ( *p*< 0.01) and a (*p* < 0.05) vs. the normal control; B (*p* < 0.01) and b (*p* < 0.05) vs. the diabetes control.

**Figure 3 foods-12-00767-f003:**
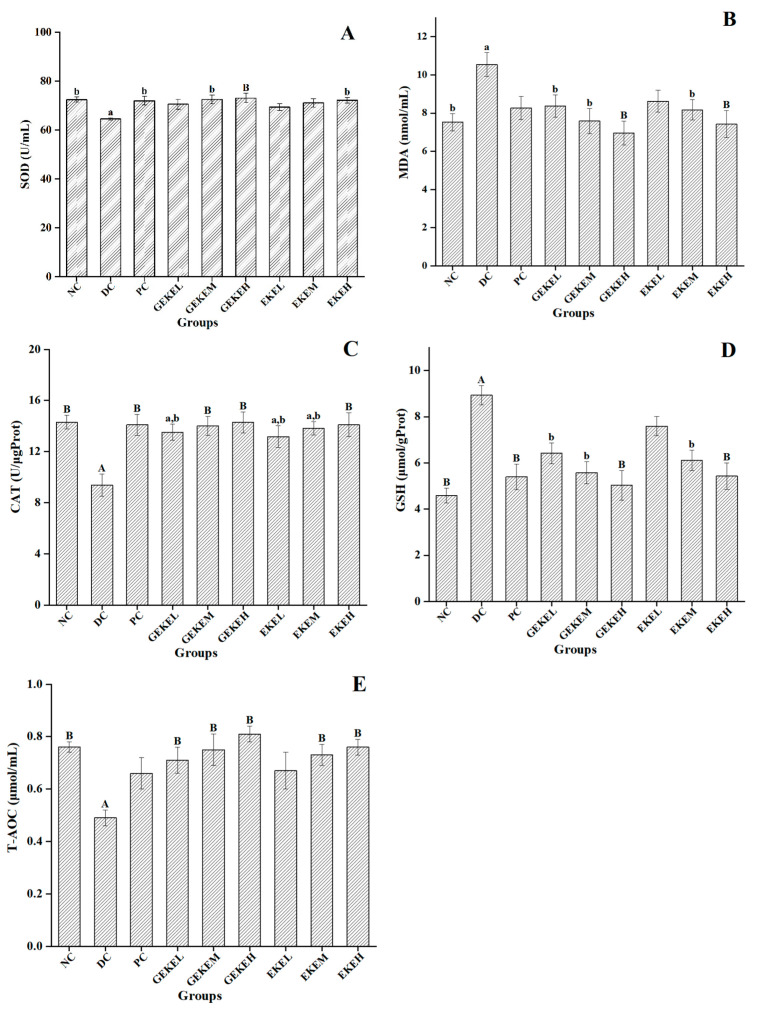
Oxidative stress parameters in each group of mice. (**A**) SOD levels in kidney supernatant of mice in each group (n = 10 per group). (**B**) MDA levels in kidney supernatant of mice in each group (n = 10 per group). (**C**) CAT levels in kidney supernatant of mice in each group (n = 10 per group). (**D**) GSH levels in kidney supernatant of mice in each group (n = 10 per group). (**E**) T-AOC levels in serum of mice in each group (n = 10 per group). A (*p* < 0.01) and a (*p* < 0.05) indicate comparison to the normal control; B (*p* < 0.01) and b (*p* < 0.05) indicate comparison to the diabetes control.

**Figure 4 foods-12-00767-f004:**
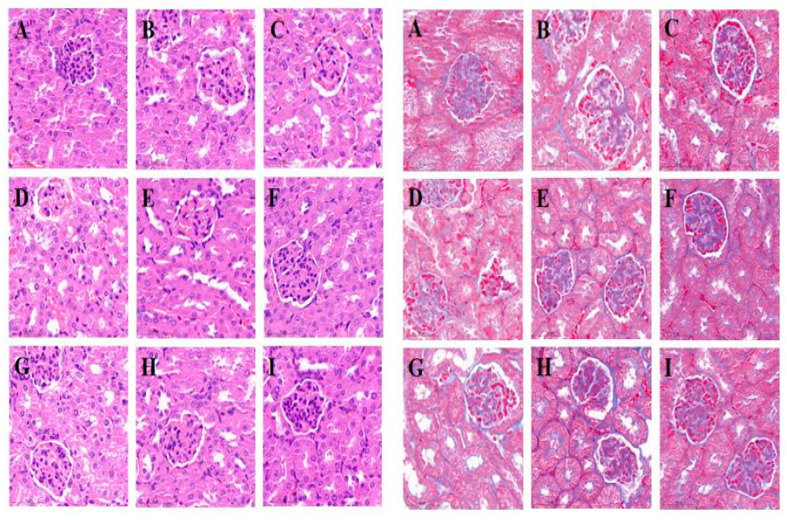
HE (**left**) and Masson (**right**) stained sections of mice kidneys. (**A**–**I**) show the kidney tissues of NC, DC, PC, GEKEL, GEKEM, GEKEH, EKEL, EKEM, and EKEH groups, respectively.

**Figure 5 foods-12-00767-f005:**
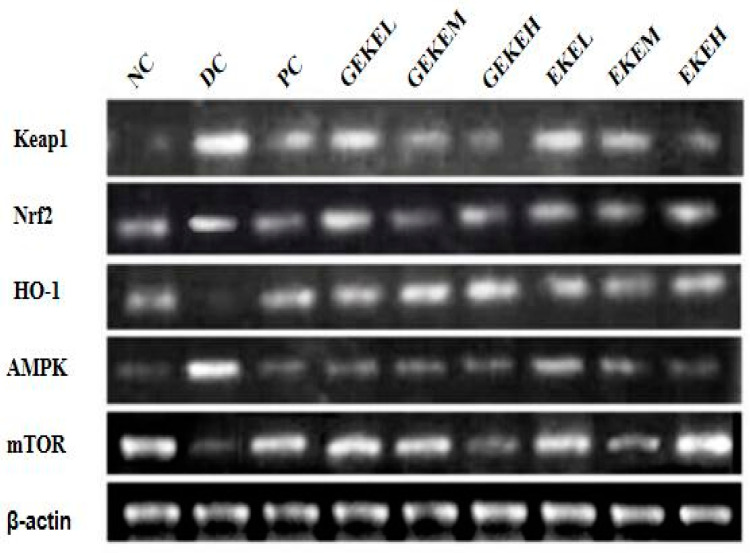
mRNA expression levels of AMPK, mTOR, HO-1, Nrf2, and Keap1 in kidney as detected by RT-PCR.

**Figure 6 foods-12-00767-f006:**
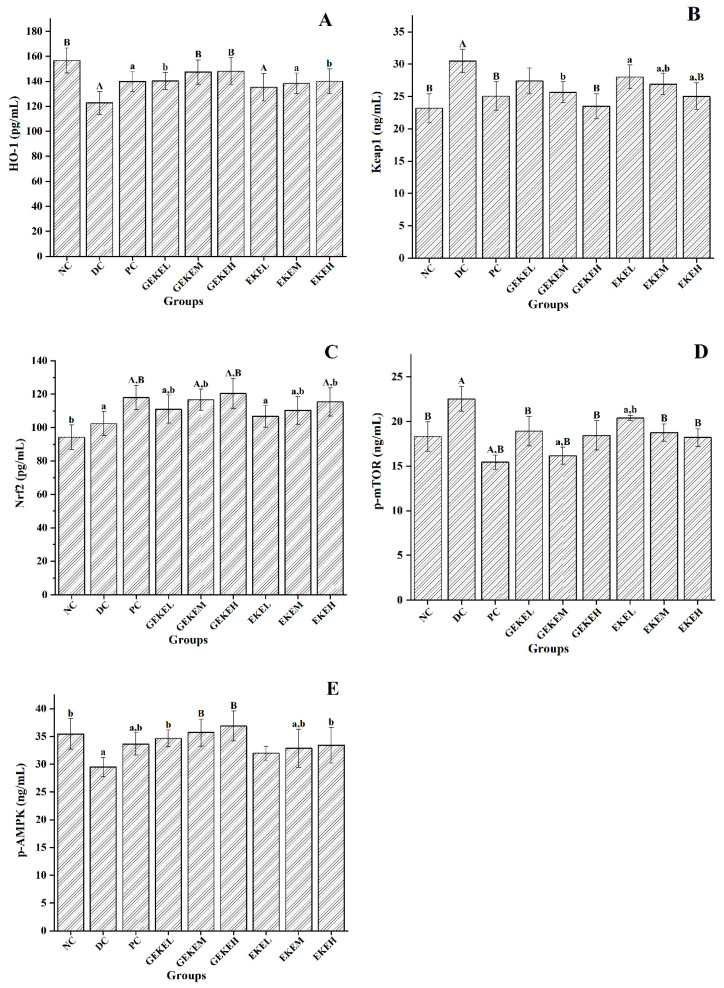
The protein expression in each group of mice. (**A**) HO-1 levels of the mice. (**B**) Keap1 levels of the mice. (**C**) Nrf2 levels of the mice. (**D**) p-mTOR levels of the mice. (**E**) p-AMPK levels of the mice. Note: n = 10 per group. A (*p* < 0.01) and a (*p* < 0.05) indicate comparison to the normal control; B (*p* < 0.01) and b (*p* < 0.05) indicate comparison to the diabetes control.

**Table 1 foods-12-00767-t001:** Treatment groups.

Group (10 Mice per Group)	STZ	Treatment ^1^	Diet
NC: Normal control group	no	None	Normal chow
DC: Diabetic control group	yes	None	Normal chow
PC: Positive control group	yes	Metformin, 200 mg/kg bw	Normal chow
EKE: Low-dose EKE group	yes	EKE, 100 mg/kg bw	Normal chow
EKEM: Medium-dose EKE group	yes	EKE, 200 mg/kg bw	Normal chow
EKEH High-dose EKE group	yes	EKE, 400 mg/kg bw	Normal chow
GEKE: Low-dose GEKE group	yes	GEKE, 100 mg/kg bw	Normal chow
GEKEM: Medium-dose GEKE group	yes	GEKE, 200 mg/kg bw	Normal chow
GEKEH: High-dose GEKE group	yes	GEKE, 400 mg/kg bw	Normal chow

^1^ Daily dose for four weeks.

**Table 2 foods-12-00767-t002:** The primers used in this study.

Gene	Primer Sequences	Amplicon Length (bp)
Nrf2	Primer F: GCTGATACTACCGCTGTTC Primer R: GTGGAGAGGATGCTGCTGA	123
HO-1	Primer F: ACACAGCACTATGTAAAGCGTCTCC Primer R: GTTGAGCAGGAAGGCGGTCTTAG	300
Keap1	Primer F: AGTGCTCAACCGCTTGCTGTATG Primer R: ACGAAAGTCCAGGTCTCTGTCTCC	243
mTOR	Primer F: ACATCACGCCCTTCACCAGTTTC Primer R:GCAGTCCGTTCCTTCTCCTTCTTG	361
AMPK	Primer F: ACCCAATTATGCCGCACCAGAAG Primer R: GGACCTCCTCCTCCGAACACTC	395
β-actin	Primer F: CACGATGGAGGGGCCGGACTCATC Primer R: TAAAGACCTCTATGCCAACACAGT	300

**Table 3 foods-12-00767-t003:** Analysis of polyphenols and terpenoids in GEKE and EKE by LC-MS/MS.

No.	Compound	RT (min)	Molecular Formula	[M+H] + (m/z)	[M-H]-(m/z)	MS/MS Fragments (m/z)	Ratio: (EKE.raw)/(GEKE.raw)
1	2-Hydroxyphenylalanine	0.719	C9 H11 N O3	165.05441		91.05470, 119.0492, 123.04409, 136.07555, 165.05446	0
2	L-Tyrosine methyl ester	1.189	C10 H13 N O3	179.0699		91.05466, 119.04919, 136.07549, 137.05949, 179.06999	0
3	2-Methoxyresorcinol	3.853	C7 H8 O3	141.05449		81.03411, 109.02865, 127.03895, 141.05444	0
4	1,2,3,4-Tetrakis-O-(3,4,5-trihydroxybenzoyl)-β-D-glucopyranose	8.721	C34 H28 O22		787.10022	125.02311, 169.01332, 787.10040	0
5	Gallic acid	4.676	C7 H6 O5		169.01326	125.02311, 169.01326	0
6	Cycloolivil	7.622	C20 H24 O7		375.14478	125.02312, 360.12140, 375.14471	0
7	2,4,6-Trihydroxy-2-(4-hydroxybenzyl)-1-benzofuran-3(2H)-one	7.183	C15 H12 O6		287.05634	125.02311, 259.06107, 287.05597	0
8	Gentisic acid	3.637	C7 H6 O4		153.0182	109.02814, 153.01825	0
9	Caffeic acid	20.522	C9 H8 O4	181.04909		149.02309, 163.03864	0
10	Matairesinol	5.895	C20 H22 O6	341.13757		137.05943, 175.07501, 309.11139	0
11	3’,4’-Dihydroxyphenylacetone	7.738	C9 H10 O3	167.0701		167.06999	0
12	Hydromorphinol	0.473	C17 H21 N O4	30416006		114.0555, 185.09174, 286.13913	0
13	(8E)-11,13-Dihydroxy-4-methyl-4,5,6,7-tetrahydro-2H-3-benzoxacyclododecine-2,10(1H)-dione	12.713	C16 H18 O5	291.12244		273.18460, 291.19470	0
14	Estriol	15.814	C18 H24 O3	311.16342		293.21042, 311.22089, 313.23651	0
15	Norbutorphanol	0.98	C16 H21 N O2	260.1601		83.06090, 84.04491, 86.09689, 242.14937, 264.14380	0
16	(-)-Fustin	4.596	C15 H12 O6	289.07028		111.04423, 123.04400, 139.03870	0
17	Resorcinol monoacetate	3.655	C8 H8 O3	153.05443		110.03642, 138.03091, 153.05434	0
18	Prostaglandin E2-1-glyceryl ester	15.041	C23 H38 O7	409.25522		409.255	0
19	Eucalyptol	17.562	C10 H18 O	137.13248		81.07042, 95.08590	Infinity
20	1,6-Bis-O-(3,4,5-trihydroxybenzoyl)hexopyranose	5.464	C20 H20 O14	502.11838		153.01787	Infinity
21	Phloroglucinol	2.236	C6 H6 O3	127.03893		130.15894, 127.03889	Infinity
22	Malvidin	13.006	C17 H14 O7	331.08066		315.04907, 331.08032	Infinity
23	7-Acetyl-3,6-dihydroxy-8-methyl-tetralone	5.475	C13 H14 O4	235.09616		129.07014, 157.06456, 217.08580, 236.09613, 239.13870	Infinity
24	3’,4’-Dihydroxyphenylacetone	7.299	C9 H10 O3	167.07011		170.11737, 167.06993	Infinity
25	Cycloolivil	9.042	C20 H24 O7	377.15891		375.14334, 377.12265, 375.14590, 379.26978	Infinity
26	2,4,6-Trihydroxyacetophenone	2.563	C8 H8 O4		167.03401	68.99425, 167.03398	Infinity
27	Octyl gallate	6.092	C15 H22 O5		281.1395	281.13943	Infinity
28	4-Methylumbelliferone	5.053	C10 H8 O3	177.05455		89.03896, 117.03363, 145.02818, 177.05418	Infinity
29	N-Acetyltyramine	4.677	C10 H13 N O2	180.10168		121.0648	0.032
30	4-Amino-3-hydroxybenzoic acid	2.614	C7 H7 N O3	154.04973		111.03170, 126.05496, 154.04962	0.164
31	N-Acetyl-L-tyrosine	3.719	C11 H13 N O4	224.09151		58.02824, 180.06584, 222.07704, 227.17508, 245.18568	0.169
32	Trolox	13.196	C14 H18 O4	251.12491		223.13126, 249.11020, 264.15900, 267.12146	0.164
33	1,6-Bis-O-[(2E)-3-(4-hydroxyphenyl)-2-propenoyl]-β-D-glucopyranose	9.115	C24 H24 O10		471.12958	125.02316, 169.01331, 471.12964, 475.03360	0.191
34	4-Acetyl-3-hydroxy-5-methylphenyl β-D-glucopyranoside	4.487	C15 H20 O8		327.10852	59.01235, 147.04401, 165.05473	0.153
35	DL-4-Hydroxyphenyllactic acid	3.152	C9 H10 O4		181.04977	135.04395, 163.03909, 181.04985	0.095
36	Epicatechin	0.908	C15 H14 O6	291.08359		291.08368, 294.15421	0.054
37	2,6-Dimethoxyphenol	6.925	C8 H10 O3	155.0701		155.06999	0.059
38	5-Hydroxyindole-3-acetic acid	5.398	C10 H9 N O3	192.06537		146.05977, 174.05470, 195.08757	0.076
39	Ethamivan	2.169	C12 H17 N O3	224.12778		160.11180, 178.12234, 224.12762	0.029
40	5-Sulfosalicylic acid	4.509	C7 H6 O6 S		216.98065	93.03311, 137.02316, 216.98061	0.028

**Table 4 foods-12-00767-t004:** Visceral indexes in each group of mice (x ± s, n = 10).

Visceral Index	Kidney (%)	Liver (%)	Spleen (%)
NC	1.35 ± 0.16 ^B^	4.49 ± 0.21 ^B^	0.32 ± 0.08 ^B^
DC	1.74 ± 0.24 ^A^	5.49 ± 0.16 ^A^	0.49 ± 0.06 ^A^
PC	1.52 ± 0.27 ^b^	4.74 ± 0.38 ^B^	0.38 ± 0.19 ^b^
GEKEL	1.49 ± 0.21 ^b^	4.69 ± 0.28 ^B^	0.36 ± 0.12 ^b^
GEKEM	1.42 ± 0.22 ^B^	4.62 ± 0.25 ^B^	0.31 ± 0.1 ^B^
GEKEH	1.41 ± 0.14 ^B^	4.51 ± 0.31 ^B^	0.36 ± 0.11 ^B^
EKEL	1.53 ± 0.18	4.76 ± 0.28 ^a,B^	0.45 ± 0.08
EKEM	1.51 ± 0.22 ^b^	4.64 ± 0.27 ^B^	0.41 ± 0.11 ^a^
EKEH	1.44 ± 0.24 ^B^	4.59 ± 0.46 ^B^	0.36 ± 0.17 ^b^

Compared with the NC group, ^A^: *p* < 0.01, ^a^: *p* < 0.05. Compared with the DC group, ^B^: *p* < 0.01, ^b^: *p* < 0.05.

**Table 5 foods-12-00767-t005:** Oral glucose tolerance test in each group of mice (x ± s, n = 10).

Group	Glucose Value/(mmol/L)					AUC/(min·mmol/L)
	0 min	30 min	60 min	90 min	120 min	
NC	5.2 ± 1.13 ^B^	13.7 ± 2.1 ^B^	10.4 ± 1.06 ^B^	5.7 ± 1.01 ^B^	3.6 ± 1.51 ^B^	1005 ± 148.69 ^B^
DC	19.8 ± 1.09 ^A^	26.00 ± 1.51 ^A^	23.2 ± 1.23 ^A^	21.2 ± 1.32 ^A^	20.3 ± 1.52 ^A^	2695.5 ± 201.88 ^A^
PC	6.8 ± 1.01 ^B^	17.8 ± 1.89	22.8 ± 1.62 ^a^	18.4 ± 1.53 ^A^	16.7 ± 1.28 ^A^	1981.5 ± 352.17 ^a,b^
GEKEL	12.9 ± 1.31	18.4 ± 1.81	23.3 ± 1.65 ^A^	21.3 ± 1.39 ^A^	19.1 ± 1.25 ^A^	2266.5 ± 532.18 ^A^
GEKEM	8.7 ± 1.18 ^b^	19.9 ± 1.46 ^a^	17.1 ± 1.38 ^a,b^	13.7 ± 1.42	10.4 ± 1.21 ^b^	1798.5 ± 561.86 ^a,b^
GEKEH	7.1 ± 1.16 ^B^	16.3 ± 1.35 ^b^	14.6 ± 1.28 ^b^	10.5 ± 1.56 ^B^	8.7 ± 1.40 ^B^	1443 ± 577.65 ^a,B^
EKEL	13.4 ± 1.21 ^a^	23.9 ± 1.68 ^A^	23.5 ± 1.46 ^A^	19.6 ± 1.52 ^A^	18.2 ± 1.26 ^A^	2431.5 ± 562.37 ^A^
EKEM	9.1 ± 1.13 ^b^	18.4 ± 1.59	16.3 ± 1.78	15.1 ± 1.23 ^A,b^	14.3 ± 1.12 ^A,b^	1858.5 ± 538.68 ^A^
EKEH	8.6 ± 1.2 ^b^	18.3 ± 1.39 ^b^	15.9 ± 1.57 ^b^	13.1 ± 1.38 ^a,b^	10.9 ± 1.51 ^b^	1705.5 ± 587.91 ^a,B^

Compared with the NC group, ^A^: *p* < 0.01, ^a^: *p* < 0.05. Compared with the DC group, ^B^: *p* < 0.01, ^b^: *p* < 0.05.

**Table 6 foods-12-00767-t006:** Serum lipid levels in each group of mice (x ± s, n = 10).

Group	T-CHO (mmol/L)	TG (mmol/L)	LDL-C (mmol/L)	HDL-C (mmol/L)
NC	5.83 ± 0.22 ^B^	1.59 ± 0.09 ^b^	0.42 ± 0.21 ^B^	3.35 ± 0.21 ^b^
DC	6.89 ± 0.2 ^A^	1.81 ± 0.1 ^a^	0.77 ± 0.24 ^A^	2.83 ± 0.45 ^a^
PC	5.77 ± 0.17	1.68 ± 0.11	0.56 ± 0.27 ^b^	3.29 ± 0.24 ^b^
GEKEL	6.73 ± 0.26 ^A^	1.63 ± 0.12 ^b^	0.61 ± 0.16 ^a^	2.96 ± 0.20
GEKEM	6.01 ± 0.21 ^a,B^	1.52 ± 0.14 ^B^	0.57 ± 0.23 ^b^	3.14 ± 0.19 ^b^
GEKEH	5.79 ± 0.17 ^B^	1.39 ± 0.12 ^A,B^	0.3 ± 0.14 ^B^	3.23 ± 0.16 ^B^
EKEL	6.79 ± 0.18 ^A^	1.67 ± 0.10	0.64 ± 0.15 ^a^	2.87 ± 0.14 ^A^
EKEM	6.35 ± 0.17	1.57 ± 0.12 ^B^	0.54 ± 0.17 ^b^	2.98 ± 0.18
EKEH	6.02 ± 0.25 ^a,B^	1.48 ± 0.13 ^a,B^	0.39 ± 0.14 ^B^	3.17 ± 0.13 ^b^

Compared with the NC group, ^A^: *p* < 0.01, ^a^: *p* < 0.05. Compared with the DC group, ^B^: *p* < 0.01, ^b^: *p* < 0.05.

**Table 7 foods-12-00767-t007:** Renal function index in each group of mice (x ± s, n = 10).

Group	ALB (μg/mL)	BUN (μg/mL)	Scr (μmol/L)
NC	21.03 ± 0.92 ^B^	11.58 ± 1.02 ^B^	36.17 ± 1.32 ^B^
DC	29.77 ± 1.12 ^A^	19.47 ± 1.27 ^A^	40.85 ± 1.04 ^A^
PC	24.41 ± 1.30 ^b^	16.03 ± 1.42^A,b^	34.17 ± 1.09 ^B^
GEKEL	24.79 ± 1.36 ^a,b^	15.11 ± 1.58	34.67 ± 1.14 ^b^
GEKEM	23.81 ± 1.21 ^B^	13.85 ± 1.33 ^B^	33.91 ± 1.01 ^B^
GEKEH	22.51 ± 1.31 ^B^	14.27 ± 1.09 ^a,B^	33.08 ± 1.03 ^a,B^
EKEL	25.41 ± 1.49	15.93 ± 1.43	38.98 ± 1.11
EKEM	25.15 ± 1.37 ^a^	15.35 ± 1.06	35.97 ± 0.94 ^b^
EKEH	24.52 ± 1.75 ^b^	14.09 ± 1.10 ^B^	34.81 ± 1.10 ^b^

Compared with the NC group, ^A^: *p* < 0.01, ^a^: *p* < 0.05. Compared with the DC group, ^B^: *p* < 0.01, ^b^: *p* < 0.05.

## Data Availability

The data provided in this study can be provided at the request of the corresponding author. Considering the security of data, data cannot be disclosed.

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
