# Peer review of "Treatment of Diabetes Nephropathy in Mice by Germinating Seeds of Euryale ferox through Improving Oxidative Stress"

_foods, 2023, doi:10.3390/foods12040767_

Round 1

Reviewer 1 Report

In this manuscript, the authors have investigated the ameliorative effects of germinating seeds of Euryale ferox on diabetic nephropathy through oxidative stress. They found that germinated seeds was more effective than un germinated one in improving the consequences of diabetic nephropathy. Following my careful review, I can find that this study is well-conducted and organized, and the analyzed parameters are suitable for this kind of studies. The results are appropriately described, and the figures are in a good order and well-prepared. The discussion is well written and explains the findings in a good order, and the conclusion is supported by the results. Therefore, I recommend this manuscript for some minor corrections.

1-  Please make the scientific name of the plant italic in the abstract and the whole manuscript.

2- The objectives of studying should be clearly mentioned in the abstract and in the last paragraph introduction section.

3- Avoid using abbreviations in the beginning of the sentences ex: NC mice line 186.

4- Anesthesia using ether not recommended and acceptable for this type of experiment ???, please comment

5-The authors are suggested to make scoring for histopathology results, the representative images not sufficient.   

6- The rationale for selecting these dosages for the germinated seeds in this experiment ? How the authors selected these doses ?

7- Recommended to clarify how many replication of experiment were involved (‘n’ numbers) in results.

8- The authors should supplement a figure about survival rate with time curve.

9- More information about ELISA assay should be provided , kit name and company name

10-  List of abbreviations should provided in the manuscript

Author Response

Dear reviewers:

On behalf of my co authors, we are very grateful to you for giving us the opportunity to revise our manuscript. We are very grateful to the editors and reviewers for giving us the title "Treatment of diabetes nephropathy in mice by germinating seeds of Euryale ferox through improving oxidative stress" Positive and constructive comments and suggestions. These comments are valuable for the revision and improvement of our paper, and have important guiding significance for our research. We have carefully studied the opinions and made corrections, hoping to get approval. Thank you again for considering publishing our manuscript. The main corrections in the paper and the responses to the reviewers' comments are shown on the next page.

We appreciate for editors and reviewers’warm work earnestly, and hope that the correction will meet with approval. Thank you very much for your attention and consideration.

Sincerely yours,

Dr. Huaibo Yuan

PhD, Assistant professor

School of Biotechnology and Food Engineering,

Hefei University of Technology

193, Tunxi Road

Hefei, Anhui 230009, China

Respond to the reviewer's comments:

  1. Response to comment:Please make the scientific name of the plant italic in the abstract and the whole manuscript.

Response:The scientific name of plants has been revised in italics in the full text. Thank you very much for your suggestion

  1. Response to comment:The objectives of studying should be clearly mentioned in the abstract and in the last paragraph introduction section.

Response:The learning objectives of the article added to the summary and conclusion have been marked with comments, "The purpose of this study was to explore the effects of GEKE and EKE treatment on antioxidant defense and metabolic capacity of diabetes animals. " added to the summary and "The purpose of this study is to provide a new way to improve the medicinal value of Euryale ferox seeds." added to the conclusion.

  1. Response to comment:Avoid using abbreviations in the beginning of the sentences ex: NC mice line 186.

Response: The sentence acronyms have been modified as follows: Normal control group that had not received STZ (NC mice) , were continuously fed an ordinary diet without any treatment.

  1. Response to comment:Anesthesia using ether not recommended and acceptable for this type of experiment ???, please comment

Response: I am very sorry to use the ether smell to make the mice unconscious in the mouse dissection experiment for spinal devascularization. The experimental method will be revised in the future animal experiments and articles.

  1. Response to comment:The authors are suggested to make scoring for histopathology results, the representative images not sufficient.  

Response: Thank you very much for your proposal. Your proposal is very good. Considering the content of the experiment, it may not be possible to grade the organization picture. We plan to put your proposal in the next article for supplement.

  1. Response to comment:The rationale for selecting these dosages for the germinated seeds in this experiment ? How the authors selected these doses ?

Response: For the selection of dosage, refer to the relevant papers of Euryale ferox seeds, which are as follows:Any dose is lethal and toxic. 400 mg/kg body weight is safe for us. Experimental design A total of 35 male albino wistar rats were utilized and were randomly divided into 7 groups of 5 animals in each group: Group I—Normal rats (untreated with dimethylsulfoxide, [DMSO]. Group-II—Diabetic control (administered with Streptozotocin (STZ). Group-III—Diabetic control + EFx seed extract (EFx) (100 mg/kg body weight). Group-IV—Diabetic control + EFx seed extract (EFx) (200 mg/kg body weight). Group-V—Diabetic control +  EFx seed extract (EFx) (300 mg/kg body weight). Group-VI—Diabetic control + EFx seed extract (EFx) (400 mg/kg body weight). Group-VII—Diabetic control + Glibenclamide (1 mg/ kg body weight) [1]. (See References1)

  1. Response to comment:Recommended to clarify how many replication of experiment were involved (‘n’ numbers) in results.

Response:Thank you for your suggestion and we will adopt it. In chapter 2.10 of the article, the number of repeated experiments is specified, and the specific modifications are as follows:All measurements were repeated 3 times and expressed as mean ± standard deviation.

  1. Response to comment:The authors should supplement a figure about survival rate with time curve.

Response: Thank you very much for your suggestion. Your proposal is very valuable. I'm sorry that there is no way to add experiments to improve it. We agree to put your proposal into another follow-up paper to explore.

  1. Response to comment:More information about ELISA assay should be provided , kit name and company name
  2. Response: The contents of the kit have been supplemented in Chapter 2.9, as follows: The protein expression levels of Nrf2, HO-1, Keap1,p-mTOR, and p-AMPK genes in the kidney tissue were measured by corresponding ELISA kits (Jiangsu Enzyme Immunity Co. LTD). Nrf2, HO-1, Keap1, p-mTOR and p-AMPK ELISA kits used in the experiment are all 96T specifications (Jiangsu Enzyme Immunity Co. LTD) .
  3. Response to comment:List of abbreviations should provided in the manuscript

Response: Add a list of abbreviations at the end of the text, as shown in the following table:

Abbreviation

Table 8. List of English abbreviations

Full English name

abbreviation

Full English name

abbreviation

ungerminated seed extract

EKE

nuclear factor erythroid 2-related factor-2

Nrf2

germinated seed extract

GEKE

Kelch-like ECH-associated protein 1

Keap1

microalbunminuria

 ALB

AMP-activated kinase

AMPK

blood urea nitrogen

BUN

Mechanistic Target Of Rapamycin

mTOR

serum creatinine

Scr

γ-aminobutyric acid

GABA

malondialdehyde

MDA

streptozotocin

STZ

 glutathione

GSH

total phenolic content

TPC

catalase

CAT

gallic acid equivalents

EAG

superoxide dismutase

SOD

Liquid chromatography

LC

total antioxidant capacity

T-AOC

Mass spectrometry

MS

Diabetes mellitus

DM

body weight

BW

type 2 diabetes mellitus

T2D

fasting blood glucose

FBG

reactive oxygen species

ROS

serum creatinine

Scr

advanced glycosylation end products

AGEs

total cholesterol

TC

oxidase-plus 1

HO-1

 LDL-cholesterol

LDL-C

NAD(P)H:quinone oxidoreductase-1

NQO1

HDL-cholesterol

HDL-C

total superoxide dismutase

T-SOD

triglycerides

TG

glycopeptin peroxidase

GSH-Px

Respond to the reviewer's comments:

1.Response to comment:Essential citations and references are missing in many procedures of the Methodology section including - Quantification of flavonoids, determination of soluble protein content, LC-MS/MS analysis, etc. The authors are suggested to include the appropriate references for those experimental methods. References should also be included in other enzymatic determinations, Histopathological, Gene expression, etc.  

Response: Thank you for your question. 24,26-33 references are added to many procedures in the methodology section and marked in yellow in the text.

2.Response to comment:Lines 170-171: Ethical approval number should be included

Response: I'm sorry that the ethical approval number is in the process of application. We will add it to the document immediately after the approval number is applied successfully.

3.Response to comment:Conclusion should be more precise and concrete. 

Response: The conclusion part has been modified as follows: Through intragastric administration of EKE and GEKE in mice, the physiological indexes of mice were improved, and there was a certain correlation with the intragastric dose. Among them, the therapeutic effect of GEKE was better. The structure analysis of the two extracts showed that GEKE contained more caffeic acid, gentic acid and other effective substances. It can be concluded, that germination of the Gordon Euryale seeds increases the content of beneficial and effective polyphenols, so that germination improves the medicinal value to treat the oxidative stress symptoms of diabetes. This conclusion was based on experiments with methanol extracts that were richer in polyphenols following germination of the seeds. In mouse experiments the extracts were shown to improve antioxidant indexes and renal indexes. GEKE can effectively improve the anti-oxidative stress ability and renal protection ability of diabetic animals by clearing excessive ROS and activating Keap1/Nrf2/HO-1 and AMPK/mTOR signaling pathways. The purpose of this study is to provide a new way to improve the medicinal value of Euryale ferox seeds.

4.Response to comment: The whole manuscript should be checked for missing appropriate citations/references and grammatical errors.  

Response: References were added to the manuscript, and the grammar of the full text was checked and polished to make the expression more accurate and clear.

Reviewer 2 Report

The investigation is interesting and bears significance to the field. Overall the manuscript has been written well. However, a few improvements are required for the manuscript which has been mentioned below: 

1) Essential citations and references are missing in many procedures of the Methodology section including - Quantification of flavonoids, determination of soluble protein content, LC-MS/MS analysis, etc. The authors are suggested to include the appropriate references for those experimental methods. References should also be included in other enzymatic determinations, Histopathological, Gene expression, etc.  

2) Lines 170-171: Ethical approval number should be included

3)  Conclusion should be more precise and concrete. 

4) The whole manuscript should be checked for missing appropriate citations/references and grammatical errors.  

Author Response

Dear reviewers:

On behalf of my co authors, we are very grateful to you for giving us the opportunity to revise our manuscript. We are very grateful to the editors and reviewers for giving us the title "Treatment of diabetes nephropathy in mice by germinating seeds of Euryale ferox through improving oxidative stress" Positive and constructive comments and suggestions. These comments are valuable for the revision and improvement of our paper, and have important guiding significance for our research. We have carefully studied the opinions and made corrections, hoping to get approval. Thank you again for considering publishing our manuscript. The main corrections in the paper and the responses to the reviewers' comments are shown on the next page.

We appreciate for editors and reviewers’warm work earnestly, and hope that the correction will meet with approval. Thank you very much for your attention and consideration.

Sincerely yours,

Dr. Huaibo Yuan

PhD, Assistant professor

School of Biotechnology and Food Engineering,

Hefei University of Technology

193, Tunxi Road

Hefei, Anhui 230009, China

Respond to the reviewer's comments:

1.Response to comment:Essential citations and references are missing in many procedures of the Methodology section including - Quantification of flavonoids, determination of soluble protein content, LC-MS/MS analysis, etc. The authors are suggested to include the appropriate references for those experimental methods. References should also be included in other enzymatic determinations, Histopathological, Gene expression, etc.  

Response: Thank you for your question. 24,26-33 references are added to many procedures in the methodology section and marked in yellow in the text.

2.Response to comment:Lines 170-171: Ethical approval number should be included

Response: I'm sorry that the ethical approval number is in the process of application. We will add it to the document immediately after the approval number is applied successfully.

3.Response to comment:Conclusion should be more precise and concrete. 

Response: The conclusion part has been modified as follows: Through intragastric administration of EKE and GEKE in mice, the physiological indexes of mice were improved, and there was a certain correlation with the intragastric dose. Among them, the therapeutic effect of GEKE was better. The structure analysis of the two extracts showed that GEKE contained more caffeic acid, gentic acid and other effective substances. It can be concluded, that germination of the Gordon Euryale seeds increases the content of beneficial and effective polyphenols, so that germination improves the medicinal value to treat the oxidative stress symptoms of diabetes. This conclusion was based on experiments with methanol extracts that were richer in polyphenols following germination of the seeds. In mouse experiments the extracts were shown to improve antioxidant indexes and renal indexes. GEKE can effectively improve the anti-oxidative stress ability and renal protection ability of diabetic animals by clearing excessive ROS and activating Keap1/Nrf2/HO-1 and AMPK/mTOR signaling pathways. The purpose of this study is to provide a new way to improve the medicinal value of Euryale ferox seeds.

4.Response to comment: The whole manuscript should be checked for missing appropriate citations/references and grammatical errors.  

Response: References were added to the manuscript, and the grammar of the full text was checked and polished to make the expression more accurate and clear.